## [Transparent Peer Review file · Nature Communications]

A Synthetic System for RNA-responsive Pyroptosis Based on Type III-E CRISPR Nuclease-Protease

Corresponding Author: Professor Mingzhou Chen

Version 0:

Reviewer comments:

Reviewer #1

(Remarks to the Author)

A Synthetic System for RNA-Responsive Pyroptosis Based on Type III-E CRISPR Nuclease-Protease

This work by He and colleagues engineers the type III-E CRISPR protease system containing the RNA-guided RNA nuclease Cas7-11 and the CHAT family protease Csx29. Normally, this protease cleaves Csx30, and the main finding of this work is that the authors engineer pore-forming gasdermin proteins to be responsive to Csx30 cleavage. This protein engineering allows for the induction of pyroptosis and cell death following recognition of target RNA programmed by the Cas7-11 guide. The authors measure this response through several cell viability assays, including PI staining and lactate dehydrogenase release, to quantify perforated cells and compare non-target and targeting conditions.

The authors test numerous gasdermin genes to make them responsive to cleavage by the Csx29 protease by inserting a Csx30 linker, forming the basic components of the DAMAGE system. This system is then applied to detect different RNA from RSV (Fig. 2), HPV (Fig. 3), KRAS (Fig. 4), and miRNA (Fig. 5). This work aims to tackle the biological problem of how one might use RNA signatures to control specific cells, here to induce cell death in response to specific RNA, which is an interesting and important problem.

I commend the authors on their substantial efforts in testing these constructs and attempting to develop a new tool. However, I am not convinced that the system works well enough. Many of these components have been used before, and this work does not adequately address the critical issues of using the CRISPR protease system in human cells, particularly regarding activity and specificity, especially in response to endogenously expressed genes.

I have two primary concerns with this work.

The first is an issue with the presentation and clarity of the paper. I found the manuscript difficult to follow, and I believe that too much unnecessary data is presented, which obscures the main findings. This is a case where “less is more,” and the authors could have selected the best DAMAGE construct from Fig. 1 and presented it more clearly and rigorously for the rest of the paper. To me it would have been more useful to optimize one construct rather than testing all the reporters in parallel with multiple target RNAs. In my opinion, Figs. 2-6 are not that useful, as we know the system is programmable and can be applied to different targets.

I also found that the text and methods lacked sufficient detail, making some experiments unclear. It is not always explicitly stated how the experiments were conducted, whether target genes were transfected or integrated, which promoters were used, and whether targets were overexpressed. Additionally, certain figure abbreviations do not seem to be explained in the text or legends.

The second major concern is the practical utility of this system outside of overexpression experiments in HEK293 cells. Many core ideas of this paper—using the CRISPR protease in human cells, truncating Csx30 as a linker to trigger biological activities, and engineering gasdermins for cleavage by other proteases—have already been explored. While this work extends these concepts in an interesting direction, it does not fundamentally advance our understanding of CRISPR protease biology or inducible cell death circuits.

I therefore disagree with the claim that this work demonstrates “considerable promise” for therapeutic applications. The PI-positive cell quantification in Fig. 1c shows an increase from 5% to 15-20%, and the ATP-based readout in Fig. 1e shows viability decreasing from 85-90% to 60-70%. As such, it seems to fall into a “proof of principle” zone but I am not sure this is sufficient for publication. In my opinion the authors would need to improve the system and show more compelling results beyond HEK293 cells.

Overall, while the study presents a potentially interesting application of previous work and attempts to tackle an important biological problem, to me it does not sufficiently address key challenges or demonstrate a substantial advance.

Specific Comments:

- The figure legends and text lack clarity in explaining the experiments and different conditions tested. For instance, the abbreviations in Fig. 1 (N- and N+) are not clearly defined. Some of the data presented feels a bit unnecessary and makes the paper “heavy” and could be moved to supplementary materials or omitted for ease of reading.
- Line 134: “All GSDMs-Csx30 effectors can respond to tgRNA and induce pyroptosis, but they display a distinct control over pyroptotic activity.” However, D-SL and D-FL clearly do not respond to target RNA. Why do the authors make this claim and continue to test these constructs that do not seem promising for this application?
- There appears to be selective presentation of different constructs across figures, which introduces confusion. For example, Fig. 1f shows C-FL, Fig. 1g shows A-FL, and Fig. 2b switches to D-X. A more consistent focus on the best reporter construct would improve the manuscript.
- The positive control in Fig. 1e at ~48% is unclear. Additionally, where are the error bars indicating standard deviation between replicates?
- A critical issue with CRISPR protease systems, as seen with other RNA-targeting systems (e.g., RADAR, CellREADR), is the requirement for high levels of target RNA. The authors should test a range of endogenous target genes to evaluate DAMAGE efficiency across physiologically relevant RNA levels.
- Fig. 6 claims LNP delivery of DAMAGE mRNA, but only Figs. 6h and 6i seem to include actual mRNA delivery. Since these experiments are performed in HEK293 cells, likely overexpressing the targets, they don’t seem to add much value beyond the plasmid transfection experiments.
- The term “tgRNA” for target RNA feels unnecessary and complicates readability.
- Some schematics, such as those in Fig. 1a and Fig. 4a, are too small. A larger, clearer depiction of a representative construct (such as crRNA-23 in the KRAS experiment) would be preferable.
- Line 89: truncation analysis of Csx30 has already been performed in previous studies and should not be claimed as novel.
- Certain panels, such as those in Fig. 1f and 1g, should directly indicate which DAMAGE constructs are being tested to improve clarity.
- Some western blot results are unclear. In Fig. 1b and 2b, the full-length and cleaved products of D-X are visible, but in Fig. 3b, the cleaved product is absent. The separate panel showing D-X-N raises questions; why is the cleaved product not observed in the FLAG blot, and what does D-X-N represent?
- In Fig. 3f and Extended Fig. 13c, why does Cas7-11-HA expression depend on crRNA? Were these experiments performed without crRNA rather than using a non-targeting control?

Reviewer #2

(Remarks to the Author)

Type III CRISPR-Cas systems provide adaptive immunity against foreign genetic elements. In the type III-E system, the Cas7-11-crRNA-Csx29 complex recognizes and cleaves RNA targets that are complementary to the crRNA guide. In addition, the binding of target RNA activates the protease activity of Csx29, leading to the cleavage of the Csx30 protein and resulting in abortive infection. In this study, He et al. integrated gasdermin proteins with the type III-E CRISPR system to develop the DAMAGE (death manipulation gene) system, which triggers pyroptosis in human cells expressing specific target RNA. Furthermore, the authors demonstrated that DAMAGE can selectively eliminate virus-infected, cancerous, and senescent cells, which exhibit altered RNA transcriptomes. The DAMAGE system has the potential to serve as a new technology for the programmable induction of pyroptosis. However, the quality of the manuscript, particularly the figures, is too poor to adequately understand the significance of the work. Therefore, I cannot support its publication in Nature Communications.

Numerous points need to be corrected, including the following:

P4: “Cas7-11 specifically recognizes target RNA (tgRNA) that is complementary to CRISPR RNA (crRNA; CR).”

- The abbreviation CR is defined for crRNA, but it is not mentioned afterward.

P7: “Both forms exhibited significantly increased in the ON+ groups (Fig. 1b).”

- This sentence is grammatically incorrect.

P7: “Meanwhile, the ON+ groups showed decreased green fluorescence intensity (Fig. 1c and Extended Data Fig. 4d), increased PI-labeled dead cells (Fig. 1c)...”

- The abbreviation PI (propidium iodide) should be defined here.

Unfortunately, the resolutions of all the figure panels and font sizes are too low to be understood. As stated in the journal

guideline, resolutions should be ~300 dpi and font sizes should be at least 5 pt (typically 7 pt). In most graphs, replicates are not provided in the legends. All data points should be plotted in the graphs. Bars should not overlap and should be presented separately (e.g., Fig. 1e).

Reviewer #3

(Remarks to the Author)

Pyroptosis is a highly inflammatory form of programmed cell death (PCD) that stands apart from other types of cell death due to its dramatic mechanism. Unlike conventional cell death processes, pyroptosis leads to the rupture of the cell membrane, releasing cellular contents and triggering a robust immune response. This process is predominantly regulated by Gasdermin proteins, with Gasdermin D (GSDMD) playing a key role. Upon detection of pathogens or danger signals by intracellular immune sensors, such as the inflammasome, enzymes like caspase-1 are activated. These enzymes cleave Gasdermin proteins, unleashing their pore-forming activity. The cleaved Gasdermin proteins then form pores in the cell membrane, allowing water to rush in. This influx causes the cell to swell and ultimately burst, releasing its contents—including inflammatory cytokines—that signal the immune system and help defend the body against infections. While pyroptosis is essential for immune defense, its excessive or inappropriate activation can be detrimental. It has been implicated in a range of diseases, including inflammatory bowel disease (IBD), sepsis, and various cancers.

In the submitted manuscript by Mingbin He et al., the authors introduce an innovative synthetic biology platform, DAMAGE, which creatively integrates CRISPR Type III-E systems with Gasdermins to induce pyroptosis. Based on their findings, I believe this represents a significant advancement in the field of programmable cell death technology. I have the following questions/comments:

Major points:

The cell lines used are relatively limited, seemingly only HEK293T and HeLa cells, which does not sufficiently demonstrate that the method works in other cell types. Therefore, it cannot be considered a universally applicable approach. The authors should validate their technology in additional cell lines and/or primary mouse cell lines. It is not necessary to repeat all similar experiments; for example, performing PI staining or LDH release in different cell lines or even mouse primary cells would be sufficient for broader validation.

1. Figure 2: Since RSV primarily infects respiratory epithelial cells, it would strengthen the physiological relevance of your findings to demonstrate the results using a respiratory epithelial cell line.
2. Figure 4: To support the generalizability of your findings, please consider including additional cancer cell lines. It may not be necessary to repeat all experiments—perhaps just the PI and LDH assays would suffice.
3. Figure 5: Can the DAMAGE selectivity also target other types of senescent cells, such as those induced by doxorubicin or H₂O₂?
4. Figure 6: Can this technology be used on other cell lines or in mouse primary cells? Expanding the validation would enhance the robustness of the findings.

Minor points

1. Figure 1c: Was the PI assay performed with three independent replicates? If not, please repeat the experiment to ensure statistical validity. Also, please clarify the meaning of “N-” and “N+” in the figure legend.
2. Figure 1: It appears that D-FL and D-SL did not work. Please explain or discuss these observations.
3. Figure 1d: Was this experiment performed with three independent replicates? If not, please do so to support the reliability of the data.
4. Figure 1e: Was this also a three-independent assay? Additionally, there is no N- control group shown, and the overlapping columns make it difficult to distinguish between groups. Consider displaying the data as in Figure 1c for better clarity.
5. Figures 1b, 2b, 2f: Please specify which antibodies were used for the Western blots detecting GSDMs-Csx30, GSDMs-Csx30-N, D-X, and D-X-N. Indicate these details in the figure legends.
6. Figure 1f: The zoomed-in view does not show pyroptotic bubbles. Please consider using a higher-resolution microscope to better visualize these features. In the absence of clear pyroptotic bubbles, it is inappropriate to classify the cells as pyroptotic based solely on phase-contrast images.
7. Figure 1g: Please clarify what FLAG label is and include this information in the figure legend.
8. Figure 2b: What does “d” refer to in the figure? Please define it in the legend for clarity.
9. Figure 2c: Please indicate how the immunoprecipitation (IP) was performed, including which antibody was used. This information should be included in the figure or figure legend.
10. In general, please ensure that all experiments are performed with at least three independent replicates to support reproducibility.
11. If possible, please include p-values in all relevant figures to indicate statistical significance.
12. Figures 4g and 4h: Please indicate the statistical method used to calculate the p-values. Also, review other figures to ensure this information is consistently provided.
13. The Methods section requires significant revision to improve clarity and reproducibility. More detailed descriptions will allow other researchers to replicate your findings.

Version 1:

Reviewer comments:

Reviewer #1

(Remarks to the Author)

I commend the authors for their additional work in revising their manuscript. In particular, I like the addition of the GFP/mCherry co-culture experiment, which is a nice demonstration of the DAMAGE system; I think this could have been moved to a main figure. Unfortunately, despite the point-by-point rebuttal and additional data provided, the main figures are essentially unchanged from the initial submission and do not meet the standard required for publication. This is not an issue of image resolution, but rather of what data are shown and how much is included in the figures.

Some issues are technical, with panels still being too small to be readable or useful (e.g., Figs. 1a and 4a), but the problems are broader than that. The paper takes a simple and clever idea and makes it complicated and difficult to get through.

While the rebuttal appears to address the concerns raised by me and the other reviewers, the manuscript does not. One of my primary issues was the selective use of different DAMAGE constructs in different assays and the lack of focus on the best-performing version across Figs. 2–6. The authors write, “In our subsequent experiments, we primarily focused on the two most potent GSDMs, GSDMB-Csx30 or GSDMC-Csx30 (containing B-FL, B-SL, C-FL, and C-SL) for further study.” However, the figures are mostly unchanged and still contain a variety of constructs without explanation.

Fig. 2c uses A-FL; Fig. 2d uses a large panel; Fig. 3b uses D-X; Fig. 3i uses a large panel; Fig. 5b uses D-X; Fig. 5c uses a large panel; Fig. 6b uses a large panel.

I think the authors are on to something with their approach, and I generally like the idea of this work; however, I do not support publication of this work in its current form.

Reviewer #2

(Remarks to the Author)

Although the authors have improved most figures in the revised manuscript, some remain difficult to interpret. For instance, Fig. 1a, 3g, 4f, 5g, h, EDF2f, and EDF5 contain very small labels, which hampers readability. Further enhancement of these figures is recommended to improve clarity. Based on the comments from Reviewer 1, there appear to be several concerns regarding the cellular experiments. As my expertise lies in biochemistry rather than cell biology, I defer to the other reviewers' judgment on whether all issues related to the cellular experiments are sufficiently addressed.

Reviewer #3

(Remarks to the Author)

The authors have answered my questions. Another important point is to further refine the figures, their legends, and the methods, ensuring that readers can clearly and readily understand the content and significance of each figure and its associated experiments. I don't have other questions.

Reviewer #4

(Remarks to the Author)

I have only seen the revised manuscript and I am a substitute for reviewer 1.

First, on the clarity of the manuscript. R1 mentioned that the manuscript was difficult to follow and that too much unnecessary data was included, which obscures the main findings. R2 raised similar points. I don't think this has been addressed properly, including the remark that unclarity may be due to image compression issues, which is besides the point. It feels like the authors misunderstood the comment. The manuscript still reads as cluttered, and the key results don't stand out clearly. Second, I'm not convinced by their response about the PI-positive cells and ATP-based readout. They explained it by talking about how to interpret PI staining and the rapid proliferation of HeLa cells. But if you look at Fig R4b, the difference between LDH release and PI staining is not that big. This makes the argument weak. To me, it looks like the cell death efficiency is simply low for this construct, which is not surprising considering it involves bulky proteins, target RNA recognition, and gasdermin activation. This is still an important step forward, but it feels very much like proof of principle, just as R1 pointed out.

What's missing is any discussion about how to move past proof of concept. If you think about this as a therapeutic, you'd probably need cell death efficiencies of 50 percent or more. Since these constructs would likely be delivered intratumorally, you can't assume multiple dosing like chemotherapy. You basically have one shot to kill as many cells as possible. It would be useful if the authors talked about possible ways to improve efficiency, even if just conceptually, so that the paper becomes a resource for others working in this area.

I also find the figures hard to interpret without reading the text. Even something simple, like explaining abbreviations in the figure captions, would help. Right now, it's difficult to figure out what things like N-, N+, OFF, and ON mean without digging through the main text.

Overall, I agree with the authors that this is a new concept. They've shown that you can kill cells based on an RNA signature, which they demonstrated using persistent viral infections and oncogenic mutations. Even though the efficiencies are low, it's still a meaningful step forward. That said, some of the claims are overstated. For example, line 75 says, “Overall, our study established a new method for the treatment of RNA-heterogeneous diseases.” This just is incorrect with the data they have now. There are similar overstatements throughout the manuscript, and I think these need to be toned down before

publication.

I do think this is a publishable piece of work, but I'm not fully sure it's at Nature Communications level yet without further refinement in the text. The data is there, and the authors should keep it more factual.

Version 2:

Reviewer comments:

Reviewer #1

(Remarks to the Author)

I appreciate the authors' detailed explanation of the different DAMAGE constructs and the corresponding revisions to the text to better justify their inclusion. These changes address my major concerns with the paper and provide appropriate context for readers and potential future users of this system.

Overall, the revised manuscript is substantially improved relative to the previous version and, in my opinion, is suitable for publication. I would have liked to see further simplification of the figures, as they remain somewhat data-dense and additional material could potentially be moved to the Supplementary Information; however, I defer to the other reviewers and the editor on this point, as this is largely a matter of presentation style.

Reviewer #4

(Remarks to the Author)

The authors have addressed my comments sufficiently and made adjustments to the manuscript to improve it.

Reviewer #1 (Remarks to the Author):

Title:

A Synthetic System for RNA-Responsive Pyroptosis Based on Type III-E CRISPR Nuclease-Protease

This work by He and colleagues engineers the type III-E CRISPR protease system containing the RNA-guided RNA nuclease Cas7-11 and the CHAT family protease Csx29. Normally, this protease cleaves Csx30, and the main finding of this work is that the authors engineer pore-forming gasdermin proteins to be responsive to Csx30 cleavage. This protein engineering allows for the induction of pyroptosis and cell death following recognition of target RNA programmed by the Cas7-11 guide. The authors measure this response through several cell viability assays, including PI staining and lactate dehydrogenase release, to quantify perforated cells and compare non-target and targeting conditions.

The authors test numerous gasdermin genes to make them responsive to cleavage by the Csx29 protease by inserting a Csx30 linker, forming the basic components of the DAMAGE system. This system is then applied to detect different RNA from RSV (Fig. 2), HPV (Fig. 3), KRAS (Fig. 4), and miRNA (Fig. 5). This work aims to tackle the biological problem of how one might use RNA signatures to control specific cells, here to induce cell death in response to specific RNA, which is an interesting and important problem.

I commend the authors on their substantial efforts in testing these constructs and attempting to develop a new tool. However, I am not convinced that the system works well enough. Many of these components have been used before, and this work does not adequately address the critical issues of using the CRISPR protease system in human cells, particularly regarding activity and specificity, especially in response to endogenously expressed genes.

We greatly appreciate the reviewer's favorable summary and review and have addressed all the points raised, which definitely help us to improve the manuscript substantially.

Despite the groundbreaking discovery that the Cas7-11-Csx29 complex can specifically cleaves Csx30¹, substantial barriers persist in leveraging this activity for clinically viable

RNA-targeted therapeutics. While current applications primarily exploit Cas7-11's nuclease activity for targeted RNA editing, our study pioneers the integration of this system with gasdermin (GSDM) family proteins. This novel strategy not only circumvents the complex activation mechanisms of the GSDM family under natural conditions, but also converts CRISPR-detected RNA signals into protease-mediated cascades, enabling precise pyroptotic cell death for treating RNA-associated pathologies.

Thanks for your suggestions regarding the application of CRISPR systems in human cells. In the final section of our study, we demonstrated that the system can be in vitro transcribed into RNA and successfully transfected into cells while retaining functionality. Furthermore, mRNA-LNP-mediated delivery achieved efficient transduction in human cell lines, resulting in effective clearance of target RNA-harboring cells. *Please see the response, Fig. R1.*

Fig. R1 The LDH release of mRNA transfection (a, b) or mRNA-LNP delivery (c), triggered by endogenous HPV18-E7 in HeLa cells

In response to queries on the efficacy of the DAMAGE system, we highlighted that transduction-limited cells expressing high-potency effectors, notably GSDMB-Csx30-FL and GSDMC-Csx30-FL exhibit near-universal pyroptotic death even in cases of endogenous triggering, validating the functional potency despite constrained delivery. *Please see the revised manuscript, Fig. 1c and Fig. 3g.*

The DAMAGE system exhibits single-base resolution in recognition of targeted KRAS-G12C mutations, as validated by allele-specific pyroptosis induction. Critical to this precision, spacer positioning of the discriminatory nucleotide was optimized through combinatorial screening. Results identified positions 16 and 23 (from the 3' spacer end) as optimal sites for G12C discrimination. *Please see the revised manuscript, Figure. 5b.* Additionally, we

transfected cells with either mCherry or EGFP combined with the DAMAGE system, but introduced only crRNA targeting EGFP. Upon mixing these two cell populations, we observed a significant reduction in green fluorescence compared to the control group, while red fluorescence remained unchanged. This result confirms the DAMAGE with strong selective specificity. *Please see the response, Fig. R2.*

Fig. R2 The co-culture experiment to verify the specificity of DAMAGE in killing target cells

We appreciate your inquiry regarding endogenous triggering capability. To address this concern, we conducted comprehensive validation at the following endogenously expressed genes:

- 1) Endogenous triggering of HPV18 E6/E7 oncoproteins in HeLa cells. *Please see the revised manuscript, Fig. 3b and g.*
- 2) Endogenous activation in KRAS-G12C engineered stable HEK293T and HeLa cell lines. *Please see the revised manuscript, Fig. 4f and Supplementary Fig. 9h.*
- 3) NCI-H23 cells with endogenous G12C mutations and A549 cells with endogenous G12S mutation to trigger DAMAGE-KRAS. *Please see the revised manuscript, Fig. 4j and k.*
- 4) Endogenous elevated p16/p21-mediated senescent cell death. *Please see the revised manuscript, Fig. 5g and Supplementary Fig. 11g.*

These experiments collectively substantiate the system's robust performance in response to endogenously expressed genes.

I have two primary concerns with this work.

The first is an issue with the presentation and clarity of the paper. I found the manuscript difficult to follow, and I believe that too much unnecessary data is presented, which obscures the main findings. This is a case where “less is more,” and the authors could have selected the best DAMAGE construct from Fig. 1 and presented it more clearly and rigorously for the rest of the paper. To me it would have been more useful to optimize one construct rather than testing all the reporters in parallel with multiple target RNAs. In my opinion, Figs. 2-6 are not that useful, as we know the system is programmable and can be applied to different targets.

We sincerely regret the issues with manuscript formatting and image clarity in our initial submission. Due to journal file size restrictions, necessary image compression inadvertently impeded viewing clarity, hindering your review process. In this revised manuscript, we have provided uncompressed, high-resolution versions of all figures to ensure optimal viewing quality.

We fully concur with your view that "less is more." However, at the outset of our research, it was unclear which gasdermin family protein fused with the Csx30 effector possessed the strongest pore-forming activity. Therefore, we investigated all GSDMs-Csx30 effectors and discovered that different effectors exhibit varying pyroptosis-inducing capacities. In our subsequent experiments, we primarily focused on the two most potent GSDMs, GSDMB-Csx30 or GSDMC-Csx30 (containing B-FL, B-SL, C-FL and C-SL) for further study.

In this study, we primarily aimed to investigate whether DAMAGE could effectively recognize target cell RNA and achieve the transduction of nucleic acid signals to protease signals, and whether this capability can be applied to disease treatment. Therefore, after constructing this system, we conducted investigations in different disease models (exogenous viral infection (RSV), cancers with integrated exogenous genomes (HPV), KRAS-mutant cancers (KRAS-G12C/KRAS-G12S) and senescent cells (p16 and p21), Fig. 2 to 5), demonstrating that the system possesses the capability to eliminate target cells. In Figure 6, we demonstrated that the system could be delivered into cells in the form of mRNA-LNP, opening possible avenues for the treatment of certain diseases associated with abnormal mRNA.

Nevertheless, we fully recognize the limitations of our study. Although the DAMAGE system has demonstrated a specific cytotoxic effect on target cells across multiple cell lines, its efficacy and functional behavior have not yet been evaluated in more complex biological systems, such as organoid or murine models. Therefore, the system is still in an early, exploratory phase of development. We greatly appreciate the suggestion that you gave, we can select the highest-activity effector for optimization in subsequent experiments. Meanwhile, additional *in vivo* studies are necessary to assess its translational potential and feasibility for future clinical applications.

I also found that the text and methods lacked sufficient detail, making some experiments unclear. It is not always explicitly stated how the experiments were conducted, whether target genes were transfected or integrated, which promoters were used, and whether targets were overexpressed. Additionally, certain figure abbreviations do not seem to be explained in the text or legends.

We are deeply sorry for not giving the sufficient detail of the text and methods. We strongly agree that additional details in the methods and results sections would enhance the clarity of our experimental procedures. In the revised manuscript, we have provided detailed descriptions of the figure elements and defined all relevant abbreviations in the text or figure legends.

The second major concern is the practical utility of this system outside of overexpression experiments in HEK293 cells. Many core ideas of this paper—using the CRISPR protease in human cells, truncating Csx30 as a linker to trigger biological activities, and engineering gasdermins for cleavage by other proteases—have already been explored. While this work extends these concepts in an interesting direction, it does not fundamentally advance our understanding of CRISPR protease biology or inducible cell death circuits.

Thanks for reviewer's attention on the experiments, outside the overexpression experiments in HEK293 cells, we also applied DAMAGE in some endogenous genes that mentioned above. These experiments collectively substantiate the system's robust performance in response to endogenously expressed genes.

The Cas7-11-Csx29-Csx30 system, as the first RNA-guided protease system^{1,2}, has seen its mechanistic understanding advance from structural biology³ insights to dynamic regulatory

networks, while its applications are centered on RNA diagnostics and synthetic biology tools which are primarily focused on the Cas7-11's nuclease activity. On the other hand, engineering heterologous protease cleavage systems has enabled Gasdermin-based platforms to overcome the inherent limitation of caspase dependency. This breakthrough underscores the tremendous therapeutic potential of Gasdermin proteins in treating tumors, fibrotic disorders, and infections. **Significantly, our study pioneers the use of Csx30 as a linker connecting the N- and C-terminal domains of GSDM proteins.** This innovative design capitalizes on the unique attributes of the Cas7-11-Csx29-Csx30 complex as an RNA-targeted proteolytic system, enabling specific detection of endogenous target RNAs and selective elimination of pathogenic cells, thereby providing a promising therapeutic strategy for RNA heterogeneous diseases.

I therefore disagree with the claim that this work demonstrates “considerable promise” for therapeutic applications. The PI-positive cell quantification in Fig. 1c shows an increase from 5% to 15-20%, and the ATP-based readout in Fig. 1e shows viability decreasing from 85-90% to 60-70%. As such, it seems to fall into a “proof of principle” zone but I am not sure this is sufficient for publication. In my opinion the authors would need to improve the system and show more compelling results beyond HEK293 cells.

Regarding the concern you raised about the relatively modest increase in the PI-positive rate, we propose two main explanations. Firstly, in our experiments, PI positive (PI+) rate (%) was quantified as the proportion of PI+ cells relative to the total cell population. Samples were collected at approximately 36 hours post-transfection, by which time nearly all successfully transfected cells had undergone death, consistent with the marked reduction in EGFP-positive (EGFP+) cells observed. However, due to inherent limitations in transfection efficiency and the rapid proliferation of HEK293 cells, un-transfected cells remained viable and expanded substantially. This led to a high total cell number at the time of sampling, thereby diluting the proportion of PI+ cells relative to the total population. Secondly, during the late stages of cell death, cellular disintegration generates fragmented debris. In Flow cytometry analysis, these fragments were gated out as cellular debris and excluded from the total cell count, further contributing to the apparent low PI-positive rate. To address your concerns, we collected samples before the cells were completely lysed, and the PI staining level was significantly increased, *please see the response, Fig. R3*. Additionally, due to the earlier sample collection time, the maximum EGFP+ rate also decreased.

Fig. R3 Characterization of pyroptotic activity of GSDMs-Csx30 effectors through Flow Cytometry assay, using EGFP as tgRNA

With respect to the relatively low reduction in ATP levels, two factors may be relevant. First, our system directly exploits the pore-forming activity of GSDMs function as the downstream executors of pyroptosis, which does not interfere with ATP synthesis pathways via signal transduction. Secondly, similar to the low PI-positive issue, the rapid proliferation of HEK293 cells likely counteracts the ATP reduction, as the increased number of viable un-transfected cells maintains overall ATP levels.

Meanwhile, we also measured LDH release, which increased by approximately 40%, also suggests the occurrence of intense pyroptosis. *Please see revised manuscript, Fig. 1d.* Western blot analysis of intracellular effector proteins revealed that full-length B-FL and C-FL were barely detectable in ON groups. *Please see B-FL and C-FL group in revised manuscript, Fig. 1b.*

Thank you for your comment regarding the initial limitation of our experiments to HEK293 cells. To address this concern and better demonstrate the applicability of DAMAGE system as you suggested, we have extended our investigations to additional cell models. Specifically, we have performed the core experiments and relevant supplementary assays in HeLa cells, 293T and HeLa stable cell lines, C33-A cells, A549 cells, and NCI-H23 cells harboring the KRAS-G12C mutation. We hope these expanded data will meet the expectations regarding the system's applicability.

Overall, while the study presents a potentially interesting application of previous work and attempts to tackle an important biological problem, to me it does not sufficiently address key challenges or demonstrate a substantial advance.

In our study, although the DAMAGE system still has aspects requiring improvement, it also demonstrates the capability to precisely recognize target RNA (including single-nucleotide mutations in KRAS), *please see the revised manuscript, Fig. 4b and e*; it can respond to

extremely low levels of target RNA (responding to RNA transcribed from plasmid concentrations of 2^{-7} μ g and RSV infection at an MOI of approximately 2^{-7}), *please see the revised manuscript, Supplementary Fig. 7a-e*, and effectively trigger pyroptosis. Importantly, the DAMAGE system is capable to detect HPV18 mRNA transcribed from the HeLa cell genome and inducing pyroptosis, suggesting its promising potential for application in the treatment of cervical cancer. As demonstrated in Fig. 6, the DAMAGE system targeting HPV18-E7 mRNA—whether delivered in plasmid (Fig. 6c, d), mRNA (Fig. 6e-h), or mRNA-LNP (Fig. 6i) form—exhibited the ability to specifically induce pyroptosis. *Please see the revised manuscript, Fig. 6*. These characteristics of the system provide promising therapeutic strategies for treating RNA-related diseases.

Specific Comments:

- The figure legends and text lack clarity in explaining the experiments and different conditions tested. For instance, the abbreviations in Fig. 1 (N- and N+) are not clearly defined. Some of the data presented feels a bit unnecessary and makes the paper “heavy” and could be moved to supplementary materials or omitted for ease of reading.

We are sorry for the lack of clarity in explaining the experiments and different conditions tested. Based on the principle that Csx29 cleaves Csx30 at residues 427-429, a plasmid expressing GSDMs-Csx30-N, which mimics N-terminal product generated by effector cleavage, was constructed as a positive control. For example, A-SL-N represents a fusion protein of GSDMA-N with a fragment of amino acids 407 to 429 of Csx30. "-" represents the group transfected with non-targeting crRNA (CR-NT), and "+" represents the group transfected with crRNA that can recognize the target RNA, for example EGFP-crRNA-Mix in **Figure 1c**.

In the revised manuscript, we have elaborated on the relevant experiments and specified the experimental conditions. The issues you raised regarding the abbreviations "N-" and "N+" (*please see the revised manuscript, Line 157-159*) and other ambiguous descriptions have been addressed. Following your suggestion, we have defined all relevant abbreviations in the corresponding figure legends. Thank you for your suggestions on the presentation of our paper. To better present our research, some redundant data have been deleted **or moved to the supplementary materials (eg: Supplementary Fig. 6d in revised manuscript)**. Additionally, we have reorganized the figures to ensure a logical progression throughout the manuscript.

- Line 134: “All GSDMs-Csx30 effectors can respond to tgRNA and induce pyroptosis, but they

display a distinct control over pyroptotic activity.” However, D-SL and D-FL clearly do not respond to target RNA. Why do the authors make this claim and continue to test these constructs that do not seem promising for this application?

We acknowledge that D-SL and D-FL do not perform as effectively as other GSDMs-Csx30 effectors. However, both D-SL and D-FL are capable to recognize and respond to target RNA. As shown in Fig. 1c and 1d, although significant cell death occurred in both D-SL and D-FL groups, the PI-positive cell rate and LDH levels in the untriggered condition (OFF) were statistically significantly lower than those in the triggered condition (ON). *Please see the response, Fig. R3 and revised manuscript, Fig. 1c, d.* (*P* values were calculated by two way ANOVA with Sidak's multiple comparisons test. And the *p*-values between OFF and ON of D-SL and D-FL were all less than 0.0001. (*****p* < 0.0001)).

We continued to investigate unmodified GSDMD constructs (D-FL and D-SL) for two primary reasons. On the one hand, the pronounced cytotoxicity of both D-FL and D-SL induces overt pyroptosis in transfected cells. This provides a visually distinguishable positive control that allows us to objectively evaluate experimental reliability and transfection efficiency. More importantly, D-SL and D-FL, as well as D-X and D-Y, represent two particularly intriguing contrasting combinations. In comparison with D-SL and D-FL, the D-X and D-Y constructs incorporated an additional short NES sequence at the N-terminus. *Please see the revised manuscript, Fig. 1a.* Notably, this modification was found to effectively reduce the non-specific pyroptotic activity of the GSDMD-Csx30 effectors, particularly in the case of D-X. Furthermore, as we mentioned in the manuscript, the pyroptosis-inducing capability of GSDMs-Csx30-SL was attenuated, while non-specific pyroptotic activity was significantly diminished, especially between B-SL and B-FL. These findings suggest that a moderate attenuation of the pyroptosis-inducing capacity in cleaved GSDMs-Csx30 may effectively reduce nonspecific pyroptotic events. We discussed this phenomenon in the Discussion section, *please see the revised manuscript, Line 575-592.* We believe that this information is important for readers who want to apply and further improve the DAMAGE system, and that they can gain valuable experience and inspiration.

- There appears to be selective presentation of different constructs across figures, which introduces confusion. For example, Fig. 1f shows C-FL, Fig. 1g shows A-FL, and Fig. 2b switches to D-X. A more consistent focus on the best reporter construct would improve the manuscript.

Thank you for reaching out the question on the presentation of the figures. During the initial phase of the experiment, we primarily used D-X to assess the pyroptosis-inducing capacity of the fusion proteins incorporating the Csx30 linker between GSDMs N-terminus and C-terminus. The results showed that the GSDMX-Csx30 effector can still be successfully cleaved by activated Csx29, *please see the revised manuscript, Supplementary Fig. 2b and c*. Subsequently, to comprehensively evaluate the potential of different GSDM family proteins, we tested effectors GSDMA to GSDME with Csx30 as linker, and identified GSDMB and GSDMC effectors exhibited superior activity. Therefore, to best demonstrate DAMAGE's efficacy, these effectors were predominantly used in subsequent experiments. However, the rapid cell death kinetics induced by B-FL and C-FL impeded timely capture of membrane ballooning during confocal microscopy. Consequently, for the **Fig. 1g** confocal analysis, we employed the less cytotoxic A-FL effector, which enabled timely visualization of pyroptotic ballooning dynamics.

- The positive control in Fig. 1e at ~48% is unclear. Additionally, where are the error bars indicating standard deviation between replicates?

We sincerely apologize for omitting details regarding the positive control. The positive control consisted of cells only transfected with the GSDMD-N-terminal fragment (D-N). This information has been added to the revised manuscript, *please see the revised manuscript, Line 166-167*. Additionally, we have added error bars to the figures and specified the number of biological replicates in the figure legends.

- A critical issue with CRISPR protease systems, as seen with other RNA-targeting systems (e.g., RADAR, CellREADR), is the requirement for high levels of target RNA. The authors should test a range of endogenous target genes to evaluate DAMAGE efficiency across physiologically relevant RNA levels.

We appreciate your suggestion to include endogenous activation experiments. We performed the following endogenous triggering assays:

- 1) Targeting endogenous HPV18 E6/E7 mRNA in HeLa cells, *please see the revised manuscript, Fig. 3b-g*.
- 2) Endogenous activation in NCI-H23 cells carrying the KRAS G12C mutation, *please see the revised manuscript, Fig. 4j*. Endogenous activation in A549 cells harboring the KRAS G12S mutation, *please see the revised manuscript, Fig. 4k*.

3) Endogenous triggering assays targeting p16/p21 in both 293T and HeLa cell lines, *please see the revised manuscript, Fig. 5g.*

Unlike other RNA-targeting systems, we propose that the Cas7-11-Csx29-Csx30 complex in DAMAGE functions as a reaction cascade with signal amplification. When even trace amounts of tgRNA are present in cells, they can be recognized by crRNA, which mediates the activated Cas7-11 to potently activate abundant Csx29. Once activated, Csx29 continuously cleaves the GSDMX-Csx30 effector, thereby constantly releasing its N-terminal fragment. The released N-terminal fragment possesses pore-forming functionality, ultimately mediating the occurrence of pyroptosis. Our titration experiments in RSV demonstrate the remarkable sensitivity of the DAMAGE system. We observed DAMAGE activation even at minimal stimulation levels: transfection of only 2^{-7} μ g RSV-N plasmid or infection with trace amounts of RSV virus. *Please see the revised manuscript, Supplementary Fig. 7d, e.* Endogenous assays successfully conducted across multiple cell lines as mentioned above further corroborate this conclusion. This strongly suggests that DAMAGE can detect target RNAs with high sensitivity in intact cells.

- Fig. 6 claims LNP delivery of DAMAGE mRNA, but only Figs. 6h and 6i seem to include actual mRNA delivery. Since these experiments are performed in HEK293 cells, likely overexpressing the targets, they don't seem to add much value beyond the plasmid transfection experiments.

We are grateful for your valuable suggestions regarding the limitations of the cell line selection and experimental design in mRNA delivery. We have supplemented the manuscript with mRNA transfection and mRNA-LNP delivery experiments in HEK293T cells, demonstrating that the DAMAGE system enables RNA-based therapeutic delivery. *Please see the response, Fig. R4.*

Fig. R4 The PI-positive rate and LDH release level of 293T when treated with mRNA (a) and mRNA-LNP (b), using HPV18 E7 as tgRNA

Beyond targeting exogenous RNA, we supplemented the study with RNA transfection experiments in HeLa cells. The result confirmed that endogenous HPV18 E7 mRNA can activate the DAMAGE system introduced via RNA and induce pyroptosis. Moreover, we conducted mRNA transfection and mRNA-LNP delivery experiments in HeLa cells and demonstrated that it could successfully respond to endogenous mRNA. *Please see the response, Fig. R5.*

Fig. R5 mRNA transfection (a) and mRNA-LNP delivery experiments (b) in HeLa cells, using endogenous HPV18 E7 as tgRNA

- The term “tgRNA” for target RNA feels unnecessary and complicates readability.

In accordance with the abbreviation "tgRNA" (target RNA) used by Kazuki Kato et al¹, we have adopted this terminology. Additionally, we also found that spelling out "target RNA" in figures caused uneven edges that compromised layout integrity and visual coherence.

- Some schematics, such as those in Fig. 1a and Fig. 4a, are too small. A larger, clearer depiction of a representative construct (such as crRNA-23 in the KRAS experiment) would be preferable.

We are sorry for the small size of some schematics. For the sake of image layout, we have made slight adjustments to the image sizes (including Fig. 1a and Fig. 4a). Meanwhile, we have uploaded zoomable, clear images for viewing details and have highlighted key steps or representative construct to improve depiction clarity in the revised manuscript.

- Line 89: truncation analysis of Csx30 has already been performed in previous studies and should not be claimed as novel.

We sincerely appreciate your identification of the problem in our Csx30 truncation analysis experiment. During our investigation, we referenced previous studies reporting Csx30 truncation experiments and did not present this as a novel discovery. However, to determine the **optimal**

length of Csx30 serving as the linker between GSDM-N and GSDM-C, we conducted similar truncation experiments to ensure the specificity these effectors. Our results identified **Csx30 407-565aa** as the optimal linker among the screened mutants. *Please see the revised manuscript, Line 97-105.*

- Certain panels, such as those in Fig. 1f and 1g, should directly indicate which DAMAGE constructs are being tested to improve clarity.

Thank you for your suggestion. We have provided clarifications in the figure legends. *Please see the revised manuscript, Line 167 and 174.* For Fig. 1f, we employed C-FL effector for experiment and it was a fluorescence microscopy image that does not involve antibody staining. And for Fig. 1g, we employed anti-Flag antibody to detect Flag-tagged A-FL effector. For Fig. 1f and 1g, we have labeled the tested DAMAGE construct in the figure legend.

- Some western blot results are unclear. In Fig. 1b and 2b, the full-length and cleaved products of D-X are visible, but in Fig. 3b, the cleaved product is absent. The separate panel showing D-X-N raises questions; why is the cleaved product not observed in the FLAG blot, and what does D-X-N represent?

We regret the insufficient clarity of some western blot results in the original submission. In the revised manuscript, we have provided additional clarification in the text and figure legends to facilitate interpretation of our findings. We also sincerely appreciate the questions you have raised. D-X-N represents the N-terminal fragment released after the GSDMD-X-Csx30 effector cleaved. The transfection experiments in Fig. 1b and 2b were conducted in 293T cells via exogenous mRNA, whereas Fig. 3b shows endogenous triggering in HeLa cells. Relative to 293T cells, HeLa cells display both lower transfection efficiency and weaker endogenous signal induction compared to the robust exogenous overexpression achievable in 293T cells. Consequently, no cleaved product was detected in Fig. 3b under the same exposure conditions. The D-X-N fragment shown in Fig. 3b was visualized through extended exposure time. Regarding the undetectable cleaved N-terminal products in FLAG blots, this occurs for the FLAG tag is fused to the effector's C-terminus. Thus, only the cleaved N-terminal fragment can be detected using specific endogenous antibodies (GSDMD, 39754S, CST) which binds to the N-terminal region of GSDMD.

- In Fig. 3f and Extended Fig. 13c, why does Cas7-11-HA expression depend on crRNA? Were these experiments performed without crRNA rather than using a non-targeting control?

Thank you for your questions and we sincerely apologize for the misunderstanding we may have caused. The decrease in the expression level of Cas7-11-HA is due to cell death but not depends on crRNA. After the cells undergo pyroptosis, the cell membrane ruptures and the cellular contents are released into the supernatant. In the ON group, the presence of Cas7-11 or other released proteins can be detected in the immunoprecipitated supernatant, *please see the response, Fig. R6*. We added an explanation this phenomenon in *lines 122-131* of the revised manuscript. The experiment of OFF group was performed using a non-targeting crRNA, which was described in *lines 119-120* of the revised manuscript.

Fig. R6 Immunoblotting assay of the cell lysates and supernatants-IP for A-FL effectors, using RSN-N as tgRNA

Reviewer #2 (Remarks to the Author):

Type III CRISPR-Cas systems provide adaptive immunity against foreign genetic elements. In the type III-E system, the Cas7-11-crRNA-Csx29 complex recognizes and cleaves RNA targets that are complementary to the crRNA guide. In addition, the binding of target RNA activates the protease activity of Csx29, leading to the cleavage of the Csx30 protein and resulting in abortive infection. In this study, He et al. integrated gasdermin proteins with the type III-E CRISPR system to develop the DAMAGE (death manipulation gene) system, which triggers pyroptosis in human cells expressing specific target RNA. Furthermore, the authors demonstrated that DAMAGE can selectively eliminate virus-infected, cancerous, and senescent cells, which exhibit altered RNA transcriptomes. The DAMAGE system has the potential to serve as a new technology for the programmable induction of pyroptosis. However, the quality of the manuscript, particularly the figures, is too poor to adequately understand the significance of the work. Therefore, I cannot support its publication in Nature Communications.

We sincerely appreciate your recognition of the research content presented in our manuscript. We would like to offer our earnest apologies for any difficulties encountered in reading the initial draft due to issues with clarity. In the revised manuscript, we have included high-resolution images, with the aim of addressing any inconveniences this may have caused. We kindly hope that our research will meet your approval.

Numerous points need to be corrected, including the following:

P4: “Cas7-11 specifically recognizes target RNA (tgRNA) that is complementary to CRISPR RNA (crRNA; CR).”

- The abbreviation CR is defined for crRNA, but it is not mentioned afterward.

To optimize the layout of the figures, we primarily employed the abbreviation CR during the figure preparation process and figure legend, such as illustrated in Fig. 4h and Fig. 5b, etc. *Please see the revised manuscript, Fig. 4h and 5b.*

P7: “Both forms exhibited significantly increased in the ON+ groups (Fig. 1b).”

- This sentence is grammatically incorrect.

Both forms exhibited significantly increase in the ON+ groups (Fig. 1b)

We are sorry for the mistakes, and have modified according to grammatical rules. *Please see the revised manuscript, Line 122-131.*

P7: “Meanwhile, the ON+ groups showed decreased green fluorescence intensity (Fig. 1c and Extended Data Fig. 4d), increased PI-labeled dead cells (Fig. 1c)...”

- The abbreviation PI (propidium iodide) should be defined here.

We sincerely apologize for our omission in specifying the full name of PI. We have rectified

this by adding PI (propidium iodide) in the revised manuscript (*Line 133*).

Unfortunately, the resolutions of all the figure panels and font sizes are too low to be understood. As stated in the journal guideline, resolutions should be ~300 dpi and font sizes should be at least 5 pt (typically 7 pt). In most graphs, replicates are not provided in the legends. All data points should be plotted in the graphs. Bars should not overlap and should be presented separately (e.g., Fig. 1e).

We sincerely apologize for the difficulties you encountered while reviewing the initial draft, which arose from poor image clarity and excessively small annotation font sizes. During the initial submission, the images were compressed due to file size restrictions for upload. However, in the revised manuscript, we have therefore provided high-resolution versions. Thanks for your suggestions regarding image clarity and font size, we have revised all the relevant figures to requested resolutions, and increased all undersized font size to ≥ 5 pt as requested in the revised manuscript. With respect to data reproducibility, we have defined all the biological replicates (n-values) in the figure legends, and incorporated all data points in the figure as possible. For Fig. 1e, given the large number of experimental groups, we employed a composite layout to present the data, with group names clearly labeled in the figure legend to optimize overall arrangement. *Please see the revised manuscript, Line 179-180.* We have also included the error bar for all data as per your recommendations.

Reviewer #3 (Remarks to the Author):

Pyroptosis is a highly inflammatory form of programmed cell death (PCD) that stands apart from other types of cell death due to its dramatic mechanism. Unlike conventional cell death processes, pyroptosis leads to the rupture of the cell membrane, releasing cellular contents and triggering a robust immune response. This process is predominantly regulated by Gasdermin proteins, with Gasdermin D (GSDMD) playing a key role. Upon detection of pathogens or danger signals by intracellular immune sensors, such as the inflammasome, enzymes like caspase-1 are activated. These enzymes cleave Gasdermin proteins, unleashing their pore-forming activity. The cleaved Gasdermin proteins then form pores in the cell membrane, allowing water to rush in. This influx causes the cell to swell and ultimately burst, releasing its contents—including inflammatory cytokines—that signal the immune system and help defend the body against infections. While pyroptosis is essential for immune defense, its excessive or inappropriate activation can be detrimental. It has been implicated in a range of diseases, including inflammatory bowel disease (IBD), sepsis, and various cancers.

In the submitted manuscript by Mingbin He et al., the authors introduce an innovative synthetic biology platform, DAMAGE, which creatively integrates CRISPR Type III-E systems with Gasdermins to induce pyroptosis. Based on their findings, I believe this represents a significant advancement in the field of programmable cell death technology. I have the following questions/comments:

We sincerely appreciate your meticulous review and valuable suggestions. Your insightful comments on the content of the manuscript have provided us with significant guidance for improvement. In response, we have added relevant experimental contents and made major revisions to the manuscript as recommended.

Major points:

The cell lines used are relatively limited, seemingly only HEK293T and HeLa cells, which does not sufficiently demonstrate that the method works in other cell types. Therefore, it cannot be considered a universally applicable approach. The authors should validate their technology in additional cell lines and/or primary mouse cell lines. It is not necessary to repeat all similar experiments; for example, performing PI staining or LDH release in different cell lines or even mouse primary cells would be sufficient for broader validation.

We sincerely appreciate your suggestion on validating our technology in additional cell lines. In accordance with your advice, we have supplemented relevant experiments in cell lines such as A549, C33-A, and NCI-H23. The details are provided in the following response or the referenced section of the revised manuscript.

1. **Figure 2:** Since RSV primarily infects respiratory epithelial cells, it would strengthen the

physiological relevance of your findings to demonstrate the results using a respiratory epithelial cell line.

Thank you for your suggestion on employment of the respiratory epithelial cell line to strengthen our findings. We performed infection experiments using RSV-susceptible A549 cells, with the DAMAGE system transfection after infection with RSV, *please see the response, Fig. R7*. It was observed that morphological apoptosis occurs in A549 cells upon normal RSV infection. However, following transfection of the DAMAGE system, it can recognize the mRNA generated post-RSV infection and trigger pyroptosis in the infected cells, thereby eliminating them.

Fig. R7 Cell imaging (a) and LDH release (b) of A549 after infected with RSV and transfected with DAMAGE-RSV

2. **Figure 4:** To support the generalizability of your findings, please consider including additional cancer cell lines. It may not be necessary to repeat all experiments—perhaps just the PI and LDH assays would suffice.

Thank you for your suggestion on adding cancer cell lines. We have included the following additional cell lines. In the DAMAGE-RSV section, we supplemented the transfection and infection experiments in A549 cell line, *please see the revised manuscript, Fig. 2i, j*; In the DAMAGE-HPV section, we employed HPV18-negative C33-A cell line as a control, *please see revised manuscript, Supplementary Fig. 8h*; In the DAMAGE-KRAS section, we initially conducted a crRNA screening targeting six common KRAS G12 mutations (G12A, G12C, G12D, G12R, G12S, G12V) in the HEK293T cell line. We selected the most effective crRNA and supplemented endogenous triggering experiments using the A549 cell line harboring the KRAS-G12S mutation and the NCI-H23 cell line carrying the KRAS G12C mutation, *please see the revised manuscript Fig. 4j, k*. PI staining, LDH release assays, or cell imaging were

performed on the referenced cell lines.

3. **Figure 5:** Can the DAMAGE selectivity also target other types of senescent cells, such as those induced by doxorubicin or H₂O₂?

Thank you for your question on the DAMAGE selectivity. In this study, we targeted the elevated transcriptional levels of p16 and p21 characteristic of senescent cells. In principle, any intervention capable of upregulating p16 or p21 transcription could potentially activate the DAMAGE system. Following your recommendation, we treated HeLa cells with Etoposide (2 μ M) and Doxorubicin (100 nM) which act as the transcriptional activator of p16 and p21, *please see the revised manuscript, Supplementary Fig. 11g, and h*. Following the treatment, cells exhibited cellular enlargement and mitotic arrest. Notably, both drug-treated groups displayed significant enhanced pyroptosis compared to untreated group (DMSO), with the Doxorubicin (100 nM) group showing particularly pronounced effects. This was evidenced by distinct pyroptotic pore formation and increased LDH release, *please see the response, Fig. R8*.

Fig. R8 LDH release of HeLa with the treatment of Etoposide or Doxorubicin (a) and Cell imaging of HeLa with the treatment of Doxorubicin (b)

4. **Figure 6:** Can this technology be used on other cell lines or in mouse primary cells? Expanding the validation would enhance the robustness of the findings.

Thank you for your suggestion. Validation of the DAMAGE system in A549 cells and other cell lines mentioned above confirmed its robust functionality. Regrettably, due to the inherent technical challenges of transfecting primary cells, we were unable to perform corresponding experiments. However, the successful cellular delivery of the DAMAGE system via mRNA-LNP in our findings has given us partial confidence to advance DAMAGE toward application. In future studies, we aim to achieve efficient delivery in murine primary cells and relevant disease models.

Minor points

1. **Figure 1c:** Was the PI assay performed with three independent replicates? If not, please repeat the experiment to ensure statistical validity. Also, please clarify the meaning of “N-” and “N+” in the figure legend.

We sincerely regret the omission of experimental details. The PI assay was performed with three independent replicates in Fig. 1c. Following the suggestion from Reviewer 1, we have repeated the relevant experiment and presented the **PI assay data as Mean with SD** in the revised manuscript.

Based on the principle that Csx29 cleaves Csx30 at residues 427-429, a plasmid expressing GSDMs-Csx30-N, which mimics N-terminal product generated by effector cleavage, was constructed as a positive control. For example, A-SL-N represents a fusion protein of GSDMA-N with a fragment of amino acids 407 to 429 of Csx30. "-" represents the group transfected with non-targeting crRNA (CR-NT), and "+" represents the group transfected with crRNA that can recognize the target RNA, for example EGFP-crRNA-Mix in **Figure 1c**. Additionally, the definitions of N- and N+ have been explicitly defined in the figure legend, *please see revised manuscript Line 157-159*.

2. **Figure 1:** It appears that D-FL and D-SL did not work. Please explain or discuss these observations.

We acknowledge that D-SL and D-FL do not perform as effectively as other GSDMs-Csx30 effectors. However, both D-SL and D-FL are capable to recognize and respond to target RNA. As shown in Fig. 1c and 1d, although significant cell death occurred in both D-SL and D-FL groups, the PI-positive cell rate and LDH levels in the untriggered condition (OFF) were statistically significantly lower than those in the triggered condition (ON). *Please see the response Fig. R9 and revised manuscript, Fig. 1c, d.* (*P* values were calculated by two way ANOVA with Sidak's multiple comparisons test (*****p* < 0.0001)).

Fig. R9 The PI-positive and EGFP-positive rate of D-SL and D-FL, using EGFP as *tgRNA*.

We continue to investigate unmodified GSDMD constructs (D-FL and D-SL) for two primary reasons. On the one hand, the pronounced cytotoxicity of both D-FL and D-SL induces overt pyroptosis in transfected cells. This provides a visually distinguishable positive control that allows us to objectively evaluate experimental reliability and transfection efficiency. More importantly, D-SL and D-FL, as well as D-X and D-Y, represent two particularly intriguing contrasting combinations. In comparison with D-SL and D-FL, the D-X and D-Y constructs incorporated an additional short NES sequence at the N-terminus. *Please see the revised manuscript, Fig. 1a*. Notably, this modification was found to effectively reduce the non-specific pyroptotic activity of the GSDMD-Csx30 effectors, particularly in the case of D-X. Furthermore, as we mentioned in the manuscript, the pyroptosis-inducing capability of GSDMs-Csx30-SL was attenuated, while non-specific pyroptotic activity was significantly diminished, especially between B-SL and B-FL. These findings suggest that a moderate attenuation of the pyroptosis-inducing capacity in cleaved GSDMs-Csx30 may effectively reduce nonspecific pyroptotic events. We discussed this phenomenon in the Discussion section, *please see the revised manuscript, Line 575-592*. We believe that this information is important for readers who want to apply and further improve the DAMAGE system, and that they can gain valuable experience and inspiration.

3. **Figure 1d**: Was this experiment performed with three independent replicates? If not, please do so to support the reliability of the data.

We are sorry that not explained clearly about the details of the experiment. The experiment in

Figure 1d was performed with six independent replicates. Following your recommendation, we have now specified the number of experimental replicates in the figure legends. *Please see the revised manuscript, Line 176-178.*

4. **Figure 1e:** Was this also a three-independent assay? Additionally, there is no N- control group shown, and the overlapping columns make it difficult to distinguish between groups. Consider displaying the data as in Figure 1c for better clarity.

The experiment in Figure 1e was conducted with three biological replicates. Whether crRNA recognizes the target sequence, both GSDMs-N-Csx30 407-429aa can directly induce pyroptosis by permeabilizing the cell membrane. Given that N- and N+ constructs exhibit comparable cytotoxic effects (Fig. 1c, d), we only presented data for the N+ group in Fig. 1e.

Thank you for your suggestion on the presentation of Fig. 1e. However, for better optimize the layout of the figures and given the multiplicity of experimental groups, we employed a chimeric display strategy for data presentation. To enhance clarity, we provided detailed labeling of distinct groups in both the figure legend and superscripts. *Please see the revised manuscript, Line 179-180.* We have also included the error bar for the data as recommended.

5. **Figures 1b, 2b, 2f:** Please specify which antibodies were used for the Western blots detecting GSDMs-Csx30, GSDMs-Csx30-N, D-X, and D-X-N. Indicate these details in the figure legends.

The C-terminus of GSDMs-Csx30 is tagged with FLAG allowing detection using an anti-Flag antibody. However, since the tag is located at the C-terminus, detection of the cleaved N-terminus requires the use of endogenous antibodies. All endogenous antibodies against GSDM proteins employed in this study target epitopes located at the N-terminus. In Figure 1b, to simultaneously detect all the effectors and their cleaved N-terminal fragments, we diluted the endogenous antibody **anti-GSDMA/B/C/D/E** by the recommended dilution ratio according to the instructions, and followed by detection of GSDMs-Csx30 and GSDMs-Csx30-N. Similarly, D-X and D-X-N in Figure 2b were detected by the endogenous antibody anti-GSDMD (GSDMD, 39754S, CST) with its antigenic epitope located at the N-terminus. Thanks for your suggestion to indicate antibody details in the figure legends. However, considering that including all antibodies in the figure legends would make the article appear cumbersome, we indicated these details in the figure and made an explanation in the Methods section, *please see the revised manuscript Line*

831-853.

6. **Figure 1f:** The zoomed-in view does not show pyroptotic bubbles. Please consider using a higher-resolution microscope to better visualize these features. In the absence of clear pyroptotic bubbles, it is inappropriate to classify the cells as pyroptotic based solely on phase-contrast images.

We sincerely apologize for the failure to clearly display the bubbling of pyroptotic cells due to inadequate image resolution. In the revised manuscript, we have uploaded high-definition images, with arrows specifically indicating the bubbling phenomenon of the cells.

7. **Figure 1g:** Please clarify what FLAG label is and include this information in the figure legend.

We are sorry for the lack of detailed clarification. Given that the C-terminus of GSDMx-Csx30 is fused with a FLAG tag, so the effector can be detected using anti-FLAG antibody. In Figure 1g, FLAG label denotes the A-FL effector incubated with anti-Flag antibody. Following your recommendation, we have supplemented this information in the figure legend, *please see revised manuscript, Line 169-174.*

8. **Figure 2b:** What does “d” refer to in the figure? Please define it in the legend for clarity.

We sincerely apologize for our oversight in failing to provide this clarification in the figure legend. The notation "d" denotes **catalytically inactive mutants** of a nuclease-inactive Cas7-11 variant (dCas7-11) or a protease-inactive Csx29 variant (dCsx29), *please see the revised manuscript Line 82-84.* Thank you for your suggestion, we have now explicitly stated this definition in the figure legend, *please see the revised manuscript, Line 235.*

9. **Figure 2c:** Please indicate how the immunoprecipitation (IP) was performed, including which antibody was used. This information should be included in the figure or figure legend.

Given that the effector protein harbors a FLAG tag at its C-terminus, we performed immunoprecipitation (IP) of supernatant using Flag-conjugated agarose beads. HA-tagged dCas-7-11 was immunoprecipitated using HA-conjugated magnetic beads, while Myc-tagged Csx29 and RSV-N were immunoprecipitated with Myc-conjugated magnetic beads. The detailed procedure was described in **Methods section**, *please see the revised manuscript, Line 831-853.* For the detection of proteins obtained via IP, we can use tag-specific antibodies (anti-FLAG, anti-HA, anti-Myc) for the corresponding tagged proteins. Following your recommendation, we

have supplemented this information in the figure.

10. In general, please ensure that all experiments are performed with at least three independent replicates to support reproducibility.

All experiments were performed with three biological replicates. Thank you for your suggestion, we have indicated the replicate values (n-values) for all experiments in the figure legends of the revised manuscript.

11. If possible, please include p-values in all relevant figures to indicate statistical significance.

Thank you for your suggestion. In the revised manuscript, we have made every effort to present all p-values and the calculation methods in the figure legends.

12. **Figures 4g and 4h:** Please indicate the statistical method used to calculate the p-values. Also, review other figures to ensure this information is consistently provided.

Thank you for your suggestion on the details of the statistical method. *P* values in Figure 4g and 4h were calculated by one way ANOVA with Sidak's multiple comparisons test. We have checked all the images and provided the explanation of all p-values calculation method in the figure legends.

13. The **Methods** section requires significant revision to improve clarity and reproducibility. More detailed descriptions will allow other researchers to replicate your findings.

Thank you for your valuable suggestion on the Methods section. We have uploaded high-definition images and supplemented detailed descriptions of the relevant experiments in the figure legends and Methods section which may assist other researchers to understand or replicate our findings.

References:

1. Strecker J, Demircioglu FE, Li D, et al. RNA-activated protein cleavage with a CRISPR-associated endopeptidase. *Science*. **378** (6622), 874-881 (2022). <https://doi.org/10.1126/science.add7450>
2. Kazuki Kato et al. A-triggered protein cleavage and cell growth arrest by the type III-E CRISPR nuclease-protease. *Science* **378**,882-889 (2022). <https://doi.org/10.1126/science.add7347>
3. Cui N, Zhang JT, Li Z, et al. Structural basis for the non-self RNA-activated protease activity of the type III-E CRISPR nuclease-protease Craspase. *Nat Commun*. **13** (1), 7549. <https://doi.org/10.1038/s41467-022-35275-5>

Point by point Response

REVIEWER COMMENTS

Reviewer #1 (Remarks to the Author):

I commend the authors for their additional work in revising their manuscript. In particular, I like the addition of the GFP/mCherry co-culture experiment, which is a nice demonstration of the DAMAGE system; I think this could have been moved to a main figure. Unfortunately, despite the point-by-point rebuttal and additional data provided, the main figures are essentially unchanged from the initial submission and do not meet the standard required for publication. This is not an issue of image resolution, but rather of what data are shown and how much is included in the figures.

We are grateful for your positive assessment on our additional data and for your further constructive suggestions. Following your advice, we have now moved the EGFP/mCherry cell co-culture experiment (Before: Extended Data Fig. 5c) to **Fig. 1g**. Please see the response, Fig. R1.

Fig. R1 Co-culture experiment of EGFP-expressing and mCherry-expressing cells. EGFP or mCherry was transiently transfected into 293T cells, with simultaneous transfection of DAMAGE targeting EGFP mRNA. And then EGFP-expressing and mCherry-expressing cells were mixed at a 1:1 ratio and co-cultured. In the ON group, a significant reduction in green fluorescence from EGFP was observed, whereas the red fluorescence from mCherry remained largely unchanged. Scale bar, 200 μm.

Besides, in the first round of review comments, you mentioned “The figure legends and text lack clarity in explaining the experiments and different conditions tested. For instance, the abbreviations in Fig. 1 (N- and N+) are not clearly defined. Some of the data presented feels a bit unnecessary and makes the paper ‘heavy’ and could be moved to supplementary materials or omitted for ease of reading”, this is essential for further improving the manuscript. Accordingly, we have removed the data of "N-" and only retain "N+" in the new revised manuscript. The N+ groups, which employed GSDMs-Csx30-N (simulate the N-terminus of GSDMs-Csx30, which is cleaved by activated Csx29) to replace the GSDMs-Csx30 in the ON group, were used as the positive control group. The specific changes include: Fig. 1b, 1c, 2b and Extended Data Fig. 6a. For details, *please see the response, Fig. R2 and Table R1.*

Fig. R2 Characterization of pyroptotic activity of GSDMs-Csx30 effectors through Flow Cytometry analysis.

The experimental treatments of the OFF group, ON group and N+ group are shown in the following table (crRNA ×: non-target crRNA, CR-NT):

Table R1

DAMAGE-EGFP					
	dCas7-11	Csx29	GSDMs-Csx30	crRNA	EGFP
OFF	✓	✓	✓	×	✓
ON	✓	✓	✓	✓	✓
N+	✓	✓	GSDMs-Csx30-N	✓	✓

We sincerely apologize that our previous revisions did not fundamentally address your concerns and have now made further adjustments on the data for better presentation. This included removing, relocating, and supplementing data as recommended. We hope these changes could significantly improve the clarity and the

overall readability of the manuscript, which might help this work to meet the standard required for publication.

Some issues are technical, with panels still being too small to be readable or useful (e.g., Figs. 1a and 4a), but the problems are broader than that. The paper takes a simple and clever idea and makes it complicated and difficult to get through.

We sincerely thank you for your constructive suggestions and your positive feedback on our idea. Following your advice, we have adjusted **Fig. 1a** and **Fig. 4a** by highlighting the most essential information and enlarging the font size in accordance with the journal's formatting guidelines.

For **Fig 1a**, we redesigned the schematic diagrams of the 12 GSDMs-Csx30 effector proteins to emphasize the distinct sequence architecture of each component, including GSDMs-N, Csx30, and GSDMs-C. Taking D-X as an example, the sequence from the N-terminal to the C-terminal is organized as follows: a short NES sequence, GSDMD-N 1→241, Csx30 407→565, and GSDMD-C 287→484. Different colors were used to represent the different components. Through these modifications, we consider that readers can clearly understand the sequence composition of these 12 GSDMs-Csx30 effectors. *Please see the response, Fig. R3.*

Fig. R3 The schematic diagram of the DAMAGE system.

For Fig. 4a, we highlighted the important sequences through boxes of different colors, including KRAS-WT (blue), KRAS-G12C (red), as well as crRNA-16 (green), crRNA-23 (purple), and crRNA-24 (orange) that exhibited the most the outstanding performance. For other crRNAs with poor performance, we marked them in gray font, and employed the lowercase red letters a/t to mark the mutant bases of KRAS-G12C. *Please see the response, Fig. R4.*

By adopting this approach, all screening sequences can be incorporated without interfering the presentation of key information. Additionally, we have included the complete set of screening sequences in the supplementary materials in the revised manuscript. *Please see the revised manuscript, Extended Table 1.* Through these modifications, we consider that readers can fully obtain the detailed sequence information of the target RNA and crRNA we used in Fig. 4.

Fig. R4 CRISPR RNA screening of the DAMAGE-KRAS system.

While the rebuttal appears to address the concerns raised by me and the other reviewers, the manuscript does not. One of my primary issues was the selective use of different DAMAGE constructs in different assays and the lack of focus on the best-performing version across Figs. 2–6. The authors write, “In our subsequent experiments, we primarily focused on the two most potent GSDMs, GSDMB-Csx30 or GSDMC-Csx30 (containing B-FL, B-SL, C-FL, and C-SL) for further study.”

However, the figures are mostly unchanged and still contain a variety of constructs without explanation.

We sincerely apologize that the previous revised manuscript did not fully address the concerns raised by you and the other reviewers, and we also greatly appreciate your critical suggestions. In this revised manuscript, we have moved some data to the supplementary materials or removed them entirely, aiming to present the findings in a more straightforward, clear, and logically coherent manner.

Regarding your concern on the selective use of different DAMAGE constructs, we first need to point out that the 12 GSDMs-Csx30 effectors designed in this study are all execution proteins of the DAMAGE system. While there are differences line in the capability to induce pyroptosis, they all share a common underlying mechanism: they respond to target RNA and trigger pyroptosis in target cells.

In this work, it was challenging to choose one "best-performing" effector, while higher lethality often comes with increased non-specific pyroptotic effects. This distinction is particularly evident between B-FL and B-SL. Therefore, we believe that B-FL and B-SL, C-FL and C-SL, as well as A-FL, A-SL, and D-X are all strong candidates among the GSDMs-Csx30 effectors. They can be utilized for different experimental purposes to yield more robust and clearer outcomes, thereby enabling a more comprehensive characterization of the DAMAGE system's functionality and its specificity in identifying target cells. Their selection was based solely on experimental requirements, not on any intentional bias.

Screening under different target conditions revealed that GSDMB-Csx30 or GSDMC-Csx30 (including B-FL, B-SL, C-FL, and C-SL) demonstrated potent cytotoxicity. The ranking of their lethality is **B-FL > C-FL > C-SL > B-SL**, but the stronger lethality also comes with higher non-specificity. We further discuss the differences in performance among these effectors in the Discussion section. *Please see the revised manuscript, **Line 635-638*** for more details.

Therefore, in Figures 2-6, we primarily investigated the stronger GSDMB-Csx30 or GSDMC-Csx30. Following your advice, to enhance experimental coherence and narrative flow of the manuscript, we consistently employed the same type of

GSDMs-Csx30 effector for each data figure in the revised manuscript. For instance, **Fig. 2** presents data derived from B-SL, **Fig. 3**, **Fig. 4**, and **Fig. 6** were based on B-FL, and **Fig. 5** utilized C-SL. Regarding the insufficient explanations in the manuscript, we have made efforts in this revised manuscript to clarify the choice of the effectors in each relevant section as suggested.

Fig. 2c uses A-FL; Fig. 2d uses a large panel; Fig. 3b uses D-X; Fig. 3i uses a large panel; Fig. 5b uses D-X; Fig. 5c uses a large panel; Fig. 6b uses a large panel.

Thank you for pointing out this issue. In this revised manuscript, we have repositioned the figures you mentioned and made every effort provided corresponding explanations. To answer your question, we first make a simple categorization of these figures:

1. "Fig. 2c uses A-FL";
2. "Fig. 3b uses D-X; Fig. 5b uses D-X";
3. "Fig. 2d uses a large panel; Fig. 3i uses a large panel; Fig. 5c uses a large panel; Fig. 6b uses a large panel".

1. "Fig. 2c uses A-FL";

For **Fig. 2c**, the main purpose of conducting this experiment is to enrich the components of the DAMAGE system in the cell culture medium supernatant through IP, for further demonstrating the release of cellular contents resulting from pyroptosis. In the Supernatant-IP component, HA-tagged Cas7-11, Myc-tagged Csx29, full-length GSDMs-Csx30 with Flag tag and its cut form GSDMs-Csx30-C can be detected by Western Blot analysis.

We presented the results of A-FL in Fig. 2c because it is highly representative. Due to its weak lethality, the pyroptosis induced by A-FL seems to be a slow and progressive process. Therefore, when samples were collected at the same time point after transfection for Western blot assay, A-FL and its cleavage form A-FL-C were more easily detectable in both cell lysates and culture supernatants compared to other stronger effectors. Following your suggestion, to maintain consistency in the use of

effectors, we repeated the experiment using B-SL and obtained consistent results and presented as the new **Fig. 2c**. Meanwhile, we also conducted the same experiment using A-SL (*Please see the revised manuscript, Extended Data Fig. 6d*), and the results remained consistent. *Please see the response, Fig. R5*.

Fig. R5 Western blot and immunoprecipitation (IP) of supernatants assay to validate the pyroptotic activity of DAMAGE-RSV, using A-SL (a), A-FL (b), B-SL (c).

2. “Fig. 3b uses D-X; Fig. 5b uses D-X”;

For **Fig. 3b** and **Fig. 5b**. The purpose of these two experiments is to preliminarily screen crRNA for tgRNA. Fig. 3b that screening the crRNA for endogenous transcription of HPV18-E6/E7 mRNA by HeLa cell genome, was done with the aim of preliminarily confirming that the endogenous mRNA in Hela cells can indeed activate the system, and of laying the groundwork for the subsequent construction of CR-Mix. And, the previous Fig. 5b is for screening the crRNA of p16/p21 mRNA, and it has been presented in the supplementary data, please see the revised manuscript, *Extended Data Fig. 3a (Please see the response, Fig. R6a)*. As an exemplar effector, D-X is widely used in these preliminary screenings. However, an effector even if GSDMs-Csx30 effectors other than D-X are used, there would expose no significant impact on the screening results. To confirm, we repeated the experiments on all targets using B-SL (*Extended Data Fig. 3b*) and the results was consistent with the D-X screening data. *Please see the response, Fig. R6b*.

Fig. R6 CRISPR RNA screening of all target RNAs, using D-X (a), B-SL (b).

Following your advice, the previous **Fig. 3b**, which shows the endogenous triggering using D-X, was replaced by the experiment employing B-FL that obtaining consistent results. *Please see the response, Fig. R7.* We have also added a description in the main text as suggested (*Please see the revised manuscript, Line 295-299*).

Fig. R7 Screening of crRNA complementary to HPV18-E6/E7 mRNA transcribed by HeLa genome, using B-FL.

3. “Fig. 2d uses a large panel; Fig. 3i uses a large panel; Fig. 5c uses a large panel; Fig. 6b uses a large panel”.

To answer this question, it is necessary to provide the following explanation of these large panel figures.

Firstly, from the research perspective, although EGFP was employed for initial validation, we cannot conclusively confirm whether the DAMAGE system exhibits identical sensing effects on different target RNAs. Therefore, in the experimental design, we planned to screen all GSDMs-Csx30 effectors for each individual target RNA. In the new revised manuscript, to reduce reading barriers, we have relocated all these large panel data prior to the main data figures and provided relevant explanations within the article to enhance overall coherence.

Secondly, the experimental designs of these large panel data figures are not entirely uniform. Based on the activation of the DAMAGE system requires five components and the diversity of target RNA properties, we have designed distinct OFF groups for different targets. The results indicated that effective activation of the DAMAGE system occurs only when all five components are simultaneously present within a single cell. This suggests that the DAMAGE system is a controllable synthetic biology framework capable of responding to target RNAs. We have provided the detailed experimental design in the corresponding figure legends, with specific details outlined in the **Table R2 (Please see the response, Table R2)**.

		DAMAGE System					tgRNA
		dCas7-11	Csx29	GSDMs-Csx30	crRNA		
EGFP (Fig. 1)	OFF	✓	✓	✓	× (CR-NT)	✓ (plasmid)	
	ON	✓	✓	✓	✓	✓ (plasmid)	
RSV-N (Fig. 2)	OFF	✓	✓	✓	✓	× (vector)	
	ON	✓	✓	✓	✓	✓ (plasmid)	
HPV18-E6/E7 (Fig. 3)	OFF	✓	✓	✓	× (CR-NT)	endogenous mRNA	
	ON	✓	✓	✓	✓	endogenous mRNA	
KRAS-G12C (Fig. 4)	OFF	✓	✓	✓	✓	KRAS-WT	
	ON	✓	✓	✓	✓	KRAS-G12C	
p16/p21 (Fig. 5)	OFF	✓	✓	✓	× (CR-NT)	✓ (plasmid+endogenous mRNA)	
	ON	✓	✓	✓	✓	✓ (plasmid+endogenous mRNA)	

Table R2 The experimental design of the OFF group and the ON group of the DAMAGE System

Beyond the two reasons mentioned above, since RSV infection itself causes a

certain degree of cell damage, the nonspecific lethality of strong effectors combined with virus-induced damage under infection conditions, is not an outcome we intended to observe. Therefore, to comprehensively evaluate the capabilities of each effector, we included all of them in the gradient experiments (The previous Fig. 2f-h which has been moved to the Supplementary Fig. 6g, h). Based on these findings, we subsequently selected B-SL, which exhibits weaker lethality compared to B-FL, for the follow-up experiments. To directly present the key findings, only the results of gradient experiment using B-SL are shown in the main figures (Fig. 2f, 2g in the revised manuscript). *Please see the response, Fig. R8).*

Fig. R8 RSV-N plasmid gradient analysis (a) and RSV infection gradient analysis (b) of DAMAGE-RSV, using B-SL.

Fig. R9 Immunofluorescence analysis for DAMAGE-RSV in RSV-infected HeLa cells

Furthermore, to better present the findings, we have removed the original Fig. 2e to the supplementary data and replaced it with a confocal microscopy assay performed in HeLa cells under infection conditions. This additional data provides direct visual evidence of pyroptosis induced by DAMAGE activation upon RSV infection. *Please see the response, Fig. R9.*

For the previous Fig. 3h and 3i, since most of the previous experiments were conducted in HEK293T cells, an initial screening was performed to evaluate the lethality of various effectors under endogenous triggering conditions in HeLa cells. A statement has been added in the manuscript, *please see the revised manuscript, Line 301-303.* For better presentation of the data, they have been relocated to Fig. 3c and 3d. The results turned out that B-FL and C-FL exhibited the strongest triggering capabilities under endogenous conditions which was consistent with the previous results in 293T. So, B-FL and C-FL were selected as the primary effectors for further investigation. Following your suggestion, we also have added relevant explanations in the manuscript, *please see the revised manuscript, Line 305-308.*

Meanwhile, we have also adjusted other sections of Fig. 5 to improve the presentation of the data. Such as the Extend Data Fig.10e, was adjusted and moved to Fig. 5d in revised manuscript, *please see the response, Fig. R10.*

Fig. R10 Cell imaging of the activation of DAMAGE-Aging by endogenous p16/p21 mRNA in 293T cell line

The original Fig. 5c and 5d have been adjusted to Fig. 5b and 5c as the

preliminary screening GSDMs-Csx30 effectors with p16 and p21 as targets. Furthermore, to better illustrate the impact of p16 and p21 mRNA levels on induction of DAMAGE, the drug (Etoposide and Doxorubicin function as the activator of p16/p21) treatment experiments in 293T cells were added, *please see the response, Fig. R11.*

Fig. R11 Pro-aging drugs (Etoposide and Doxorubicin) further exacerbate DAMAGE-Aging-induced pyroptosis

For **Fig. 6b**, after constructing DAMAGE-Plus, we re-conducted a preliminary investigation into the lethality of all effectors. The resulted showed that B-FL possessed the highest lethality based on the screening and was chosen for the subsequent experiments. A relevant explanation has been provided in the main text, *please see revised manuscript, Line 528-532.*

I think the authors are on to something with their approach, and I generally like the idea of this work; however, I do not support publication of this work in its current form.

We truly appreciate your recognition of our approach and idea. We are also very grateful for your constructive comments. We have carefully addressed each point in the revised manuscript and hope that these revisions could fully resolved your concerns, bringing the paper to a publishable standard.

Reviewer #2 (Remarks to the Author):

Although the authors have improved most figures in the revised manuscript, some remain difficult to interpret. For instance, Fig. 1a, 3g, 4f, 5g, h, EDF2f, and EDF5 contain very small labels, which hampers readability. Further enhancement of these figures is recommended to improve clarity. Based on the comments from Reviewer 1, there appear to be several concerns regarding the cellular experiments. As my expertise lies in biochemistry rather than cell biology, I defer to the other reviewers' judgment on whether all issues related to the cellular experiments are sufficiently addressed.

We sincerely thank you for your suggestion. Following your advice, we have maximized the font size of the scale bars in Fig. 1a, 3g, 4f, 5g, h (main figure) according to the journal's formatting guidelines. And for EDF2f, and EDF5 that contains numerous small images, we removed the labels and retained only the bars, and the specific scale values are clearly described in the figure legends. Furthermore, we have fully addressed the issues raised by the other reviewers to the best of our ability. We hope that the comprehensive revisions have now satisfactorily resolved all concerns and the manuscript meets the journal's standard for publication.

Reviewer #3 (Remarks to the Author):

The authors have answered my questions. Another important point is to further refine the figures, their legends, and the methods, ensuring that readers can clearly and readily understand the content and significance of each figure and its associated experiments. I don't have other questions.

Thank you for your valuable suggestions which has greatly helped us in enhancing this work. Following your and other reviewers' suggestions, we have further refined the data presentation in the figures and optimized several labels that were too small. We have also taken care to provide thorough descriptions of the experimental procedures in both the figure legends and the Methods section, ensuring that readers can more easily and accurately grasp the content and significance of each figure and its associated experiments. Finally, we sincerely appreciate your positive assessment of our work.

Reviewer #4 (Remarks to the Author):

I have only seen the revised manuscript and I am a substitute for reviewer 1.

Thank you for reviewing our manuscript and providing the valuable suggestions. We have revised the manuscript accordingly and hope that these changes could fully address all your concerns, thereby meeting the journal's requirements for publication.

First, on the clarity of the manuscript. R1 mentioned that the manuscript was difficult to follow and that too much unnecessary data was included, which obscures the main findings. R2 raised similar points. I don't think this has been addressed properly, including the remark that unclarity may be due to image compression issues, which is besides the point. It feels like the authors misunderstood the comment. The manuscript still reads as cluttered, and the key results don't stand out clearly.

We are grateful for your feedback and sincerely regret that our initial revisions did not adequately resolve the concerns of Reviewers 1 and 2. In response to the valuable comments from you and the other reviewers, we have refined the data presentation in the revised manuscript avoiding obscuring the main findings. These changes include:

- The removal of redundant datasets (*Please see revised manuscript, Fig. 1b, 1c, 2f and 2g etc.*).
- The relocation of less critical data to the supplementary figures (*Please see revised manuscript, Extend Data Fig. 6b, 6f, 6g and 6h etc.*).
- Adjustments to specific panels in the main figures (*Please see revised manuscript, Fig. 2 etc.*).
- The addition of new data to strengthen our conclusions (*Please see revised manuscript, Fig. 2c, 2d and 2e etc.*).

Furthermore, the manuscript has been extensively refined to enhance the readability and stand out the key results clearly.

Second, I'm not convinced by their response about the PI-positive cells and ATP-based readout. They explained it by talking about how to interpret PI staining and the rapid proliferation of HeLa cells. But if you look at Fig R4b, the difference between LDH release and PI staining is not that big. This makes the argument weak. To me, it looks like the cell death efficiency is simply low for this construct, which is not surprising considering it involves bulky proteins, target RNA recognition, and gasdermin activation. This is still an important step forward, but it feels very much like proof of principle, just as R1 pointed out.

Thank you very much for your positive feedback on our methodological design. We fully agree with your perspective that factors such as the large-scale protein, target RNA recognition, and the activation of gasdermin family proteins can indeed significantly influence the overall system performance. Thus, the system still faces considerable challenges and requires significant improvements before its practical application. We fully acknowledge your and Reviewer 1's perspective that our work represents a proof of principle since data in **Figure 6** provided only preliminary evidence that DAMAGE holds the potential of delivering via LNPs.

However, under the transfection conditions which ensures adequate protein expression, we are inclined to think that the pyroptosis induced by DAMAGE resembles a cascading amplification reaction. The upstream recognition of tgRNA by crRNA mediates the activation of Cas7-11—Csx29, subsequently triggering the cleavage of downstream effectors. But, the varying potency of different effectors is indeed a key factor limiting the overall lethality of the system. While, all full-length effectors (GSDMs-Csx30-FL: A-FL to E-FL) exhibited greater lethality than their shorter counterparts (GSDMs-Csx30-FL: A-SL to E-SL) (Discuss in Discussion Section, *please see revised manuscript, Line 629-649*). The experiments in this study have confirmed that transfection efficiency is a major reason for the relatively low PI staining. Even under conditions where GSDMD-Csx30-N was directly transfected to perforate cell membranes, flow cytometry detected only approximately 45% PI-positive cells. Therefore, whether in transfection experiments or the practical applications, delivery efficiency poses a significant challenge for DAMAGE.

What's missing is any discussion about how to move past proof of concept. If you think about this as a therapeutic, you'd probably need compared to the separate components.. Since these constructs would likely be delivered intratumorally, you can't assume multiple dosing like chemotherapy. You basically have one shot to kill as many cells as possible. It would be useful if the authors talked about possible ways to improve efficiency, even if just conceptually, so that the paper becomes a resource for others working in this area.

We sincerely appreciate you highlighting this limitation. Our study was primarily focused on the initial proof-of-concept for the delivery system, and we regret not having discussed its practical efficiency or potential strategies for improvement.

Indeed, we have actively explored strategies to enhance delivery efficiency. As illustrated in Figure 6, we developed the DAMAGE-Plus system by effectively linking the three protein components of the DAMAGE—Cas7-11, Csx29, and GSDMs-Csx30—using T2A and P2A self-cleaving peptides. This modification not only streamlined the DAMAGE system but also partially enhance efficient cellular delivery of the components compared to the separate components. For LNP delivery, we attempted to fuse Csx29, GSDMB-Csx30-FL into a single plasmid. This approach achieved successful delivery and subsequently induced pyroptosis (**Fig. 6g-i**). Nevertheless, these studies constitute only initial explorations and additional work is warranted to enhance the system's efficacy.

Following your insightful suggestions, we have discussed potential strategies on the possible ways to improve efficiency, *please see revised manuscript, **Line 650-658*** in **Discussion** section. We hope that the proposed concepts will serve as a valuable resource for researchers and practitioners in this field.

Meanwhile, it is noteworthy that pyroptosis is a form of inflammatory programmed cell death characterized by its ability to recruit and activate immune cells, thereby enhancing the elimination of tumor tissues. **The scenario depicted below is our most desired outcome; however, it currently remains an ambitious hypothesis that requires substantial experimental validation.** Specifically, in the context of cancer therapy, the DAMAGE system is expected to function as a

molecular trigger that induces pyroptosis through specific recognition and targeting of tumor-associated mRNAs. This process may promote immune infiltration and activation, potentially transforming immunologically "cold" tumors into "hot" tumors (*please see the response*, Reference 1 and 2 for details, and **Fig. R12**). The relevant content has also been mentioned in the **Introduction** section. *Please see the revised manuscript, Line 41-43*. Consequently, we hypothesize that achieving a tumor cell death efficiency of 50% or higher may not be strictly necessary for effective suppression of tumor progression. Nevertheless, it should be emphasized that current studies remain at the cellular level, and in vivo validation has not yet been conducted. Therefore, whether DAMAGE system-mediated pyroptosis can elicit a robust anti-tumor immune response through immune recruitment and activation warrants further investigation.

[Figure Redacted]

Fig. R12 Immunological outcomes of gasdermin pore formation and pyroptosis.

(NatRev Immunol. 2020;20(3):143-157.)

Pyroptotic cells release numerous intracellular molecules that can activate the immune system by acting as alarmins and 'find me' signals.

I also find the figures hard to interpret without reading the text. Even something simple, like explaining abbreviations in the figure captions, would help. Right now, it's difficult to figure out what things like N-, N+, OFF, and ON mean without digging through the main text.

Thanks for your suggestions, and meanwhile, to better highlight the key findings, we have removed the data of "N-" and only retain "N+" in the new revised manuscript. The N+ groups, which employed GSDMs-Csx30-N (simulate the N-terminus of GSDMs-Csx30, which is cleaved by activated Csx29) to replace the GSDMs-Csx30 in the ON group, were used as the positive control groups. The specific changes include: **Fig. 1b, 1c, 2b** and **Extended Data Fig. 6a**. For details, *please see the response, Fig. R2 and Table R1*.

Following your suggestion, we have accordingly added explanations for all relevant abbreviations and details of the experiments in each figure caption. This could facilitate easier interpretation for you and other readers.

Fig. R2 Characterization of pyroptotic activity of GSDMs-Csx30 effectors through Flow Cytometry analysis.

The experimental treatments of the OFF group, ON group and N+ group are shown in the following table (crRNA ×: non-target crRNA, CR-NT):

Table R1

DAMAGE-EGFP					
	dCas7-11	Csx29	GSDMs-Csx30	crRNA	EGFP
OFF	✓	✓	✓	×	✓
ON	✓	✓	✓	✓	✓
N+	✓	✓	GSDMs-Csx30-N	✓	✓

Overall, I agree with the authors that this is a new concept. They've shown that you can kill cells based on an RNA signature, which they demonstrated using persistent viral infections and oncogenic mutations. Even though the efficiencies are low, it's still a meaningful step forward. That said, some of the claims are overstated. For example, line 75 says, "Overall, our study established a new method for the treatment of RNA-heterogeneous diseases." This just is incorrect with the data they have now. There are similar overstatements throughout the manuscript, and I think these need to be toned down before publication.

We sincerely appreciate your encouraging feedback and insightful comments. While our study only represents a preliminary attempt to explore practical applications, we acknowledge and apologize for the inclusion of some overly assertive statements in the original manuscript. In the revised version, we have carefully toned down these overstated claims to present a more scientific explanation of the findings, ensuring they meet the standards required for publication. Line 75 that you mentioned has revised to "In summary, DAMAGE represents a sophisticated circuit that transduces signals by converting RNA information to proteolytic activity, facilitating the cascading amplification of these signals. We maintain high expectations regarding the potential applications of DAMAGE, it may offer a new therapeutic strategies for a wide range of RNA-heterogeneous diseases". The other similar overstatements were also refined, *please see revised manuscript, Line 659-663.*

I do think this is a publishable piece of work, but I'm not fully sure it's at Nature Communications level yet without further refinement in the text. The data is there, and the authors should keep it more factual.

We sincerely appreciate your positive assessment of our work and your critical suggestions. In the revised manuscript, we have performed comprehensive refinement in the text and the data presentation to enhance the scientific rigor and accuracy, ensuring that it meets the standards for publication at *Nature Communications*.

References:

- [1] Broz, P., Pelegrín, P., and Shao, F. The gasdermins, a protein family executing cell death and inflammation. *Nat. Rev. Immunol* 20, 143–157 (2020).
- [2] Liu, X., Xia, S., Zhang, Z., Wu, H., and Lieberman, J. Channelling inflammation: gasdermins in physiology and disease. *Nat. Rev. Drug Discov* 20, 384–405 (2021).

Point-by-point response to the reviewers' comments

Reviewer #1 (Remarks to the Author):

I appreciate the authors' detailed explanation of the different DAMAGE constructs and the corresponding revisions to the text to better justify their inclusion. These changes address my major concerns with the paper and provide appropriate context for readers and potential future users of this system.

We sincerely appreciate your positive feedback on our work and are truly delighted that we have been able to address all your concerns. Thanks for the time you dedicated and the invaluable suggestions you raised, which have played a significant and essential role in refining our manuscript.

Overall, the revised manuscript is substantially improved relative to the previous version and, in my opinion, is suitable for publication. I would have liked to see further simplification of the figures, as they remain somewhat data-dense and additional material could potentially be moved to the Supplementary Information; however, I defer to the other reviewers and the editor on this point, as this is largely a matter of presentation style.

Thank you for your positive assessment that our manuscript is suitable for publication. In light of the suggestions from you and other reviewers, and given the extensive scope of this work, we have strived to include only the most essential data in the main figures, while placing supplementary or more detailed elements in the Supplementary Figures. Following your advice, we have further refined the presentation of several figures in the revised manuscript (*Please see Figure. 1d*) to improve the clarity and readability. Additionally, we have carefully revised and polished the manuscript point-by-point according to the author checklist provided by the editorial office. We trust these adjustments will support the successful publication of this work.

Reviewer #4 (Remarks to the Author):

The authors have addressed my comments sufficiently and made adjustments to the manuscript to improve it.

We are very pleased to have addressed all your concerns and are grateful for your positive feedback on our work. Thank you once more for your time and the crucial advice provided during review.